# Using Quantitative Hormonal Fertility Monitors to Evaluate the Luteal Phase: Proof of Concept Case Study

**DOI:** 10.3390/medicina59010140

**Published:** 2023-01-10

**Authors:** Thomas P. Bouchard

**Affiliations:** Department of Family Medicine, University of Calgary, 108-30 Springborough Blvd SW, Calgary, AB T3H 0N9, Canada; tbouchar@ucalgary.ca

**Keywords:** luteal phase, fertility monitor, pregnanediol glucuronide (PDG), luteinizing hormone (LH)

## Abstract

Several new quantitative fertility monitors are now available for at-home use that measure estrogen, luteinizing hormone (LH), and progesterone (PDG) in urine. This case report compares the Mira and Inito quantitative fertility monitors with the well-established qualitative ClearBlue fertility monitor. Three clinical scenarios were evaluated: a normal cycle, a prolonged luteinization cycle, and an anovulatory cycle. The identification of the luteal phase (or lack thereof in the case of anovulation) and the transition through the three processes of luteinization, progestation, and luteolysis were clearly demarcated with the help of quantitative LH and PDG. Quantitative fertility monitors have the potential to identify details of the luteal phase to help women with regular cycles and abnormal luteal phases to help target interventions for optimizing fertility.

## 1. Introduction

The quantitative self-monitoring of urinary hormones is a rapidly advancing field of precision medicine for reproductive health. A recent study showed a high degree of correlation between an existing qualitative monitor (ClearBlue fertility monitor, CBFM) and a new quantitative fertility monitor (Mira monitor) for identifying the fertile window [1]. Both the CBFM and Mira measure estrone-3-glucuronide (E3G), the urinary metabolite of estrogen, and luteinizing hormone (LH) in the urine; Mira has also recently added a progesterone test, pregnanediol glucuronide (PDG), and a follicle-stimulating hormone (FSH) test to their system. In addition to the Mira monitor, there are now multiple monitors available for identifying quantitative changes in urinary hormones. The Inito monitor (inito.com) and the Proov testing system (proovtest.com) both measure E3G, LH, PDG, and FSH. Another quantitative monitor, Oova (oova.com), measures LH and PDG. All four of these quantitative testing systems (Mira, Inito, Proov, and Oova) include a synced smartphone app that graphically displays the quantitative hormone levels and provides predictions regarding the fertile window for the user. 

Based on a review of the literature as well as correspondence with the manufacturers of these new systems, there are very few published studies validating the clinical performance and accuracy of these monitors. Besides the comparison between Mira and the CBFM [1], there are two studies validating the Inito monitor [2,3], and one study in-press on the new Proov system, but there are no published studies on the Oova monitor. There are a few studies that previously evaluated the original Proov progesterone tests, but these were not quantitative tests [4,5]. In this rapidly evolving field, there is an urgent need for studies to evaluate these new quantitative monitors for users to have confidence that the data being provided by the monitors are clinically accurate and reliable.

The identification of ovulation using luteinizing hormone is well established, and there are different devices and test sticks for this purpose [6,7], but certain thresholds are likely better than others for detecting ovulation [8]. Using LH to detect ovulation permits the calculation of the luteal phase length by subtracting the cycle length from the estimated day of ovulation but does not give any details about the progesterone dynamics in the luteal phase. In a European dataset, a more detailed analysis of hormone profiles during the luteal phase showed differences between women and between cycles that can be characterized by three different processes: first, the formation of the corpus luteum (luteinization) with the interaction of LH and PDG; second, the progestation process, when PDG rises to support a potential pregnancy; and third, the luteolysis process, when the corpus luteum regresses as PDG levels decline [9]. To achieve this level of detail, a PDG measurement is required.

The development of an accurate urinary PDG test first required establishing a threshold that would confirm ovulation that was referenced to ultrasound [10]. Using two different thresholds (5 and 7 µg/mL), a progesterone test stick (proovtest.com) was developed and tested in two different populations of women [4,5], with a higher rate of ovulation confirmation with the 5 µg/mL test sticks (82%) compared to the 7 µg/mL test sticks (59%) [5]. However, these line-based tests do not provide the detailed information needed to classify the dynamic changes in the luteal phase that would assist with distinguishing normal from abnormal luteal phases, especially for the purposes of optimizing conception. 

The identification of a “deficient” luteal phase has been the subject of some controversy [11], mainly because the traditional evaluation of progesterone is conducted with a “day 21” progesterone test, which assumes ovulation always happens on cycle day 14, which does not reflect the normal variations in women’s cycles [12]. Other measures of the luteal phase (luteal phase length of 12–14 days and basal body temperature measurements) also lack precision. Despite the imprecision of these measurements, the evaluation of the luteal phase remains an important biomarker of fertility [13], and many studies have already been carried out to attempt to supplement progesterone in the case of recurrent miscarriage [14,15]. The greatest benefit of progesterone for recurrent miscarriage is during the luteal phase [15], rather than after a positive pregnancy test [16], which further highlights the importance of identifying abnormal luteal phases and the early timing of progesterone supplementation.

Specific patterns have been identified that predict abnormalities in the luteal phase. For example, low PDG around the time of ovulation predicts low PDG in the mid-luteal phase [17]. Other details related to a shortened luteal phase length, estrogen levels, LH, and follicle-stimulating hormone (FSH) were also found to be relevant with respect to luteal phase deficiency [18,19]. 

The advent of quantitative fertility monitors measuring progesterone will hopefully allow for more precise measurement in order to identify abnormalities and provide the foundation for clinical interventions that could correct these abnormalities. The current case study evaluated three clinical scenarios (a normal cycle, a late and broad LH surge with delayed ovulation, and an anovulatory cycle) using two quantitative monitors (Mira and Inito) measuring E3G, LH, and PDG compared to a qualitative monitor (CBFM) measuring E3G and LH. This study focused on identifying different patterns in LH leading up to the luteal phase and PDG patterns during the luteal phase.

## 2. Materials and Methods

A single participant provided daily first morning urine samples starting on day 6 of the cycle. Urine hormones were analyzed with the CBFM and Mira monitors (described in detail in a previous study [1]) as well as with the Inito monitor (inito.com) using lateral flow assays (a sandwich assay for LH and a competitive assay for E3G). The CBFM measures E3G and LH on a single test stick, and the monitor provides “Low”, “High”, and “Peak” results, where “High” represents a rise in E3G and “Peak” represents passing the threshold level of LH (although this is proprietary information, “High” values are likely around 200 ng/mL of E3G, and “Peak” values are typically >30 mIU/mL of LH). The Mira monitor measures E3G and LH on a single test stick and PDG on a separate test stick and syncs via Bluetooth to the Mira App, which graphically displays the results. The Inito monitor mounts onto a smartphone with a clip customized to the phone and uses the phone’s camera with a controlled light in the device to measure the intensity of the lines on a single test strip measuring E3G, LH, and PDG (unpublished manufacturer data). 

The estimated day of ovulation (EDO) was determined based on previously established criteria for CBFM and Mira monitor [1], and the Inito results were compared to the EDO on these two monitors. The EDO for the CBFM and Mira monitor occurs most often on the day after the peak [1]. Despite there being established thresholds for urinary hormones [20], these new monitors have not yet been referenced to these ranges, and thus progesterone patterns were followed without having a specific threshold in mind, as was the case with the original Proov test thresholds [4,5]. 

## 3. Case Presentation

A 39-year-old G4P4 (BMI 24.0, otherwise healthy with no specific medical conditions) woman recorded six menstrual cycles with all three fertility monitors. The average cycle length was 27 days (range 25–34 days), with an average peak day (first peak on CBFM) of 12.6 (range 11–19 days) and an average luteal length of 13.4 days (range 13–14 days). 

### 3.1. Normal Cycle

All three monitors demonstrated a typical pattern that would be expected for a normal cycle, with a clear LH surge and a rise in progesterone that happened a few days later (Figure 1). On cycle day 11, the LH peak on the Inito monitor was 40 mIU/mL and the LH peak on the Mira monitor was 57 mIU/mL. This peak and the following reduction in LH delineates the luteinization process. The PDG plateau (i.e., demonstrating the progestation process) was 14 ug/mL on the Inito monitor and 15 ug/mL on the Mira monitor (this was the PDG ceiling for each monitor). PDG declined abruptly (i.e., demonstrating the luteolysis process) on both monitors on day 22, and the cycle was complete 3 days later (25-day cycle).

### 3.2. Prolonged Luteinization Process

In this longer cycle (34 days), there was a broader LH surge, with a slightly different progesterone plateau (Figure 2). On cycle day 19, the LH peak on the Inito monitor was 40 mIU/mL, and the LH peak on the Mira monitor was 75 mIU/mL, but the LH levels remained high until about day 22, demonstrating a prolonged luteinization process. The PDG rise for the Inito monitor was on day 22, and the PDG rise for the Mira monitor was on day 23. The PDG plateau was 12 ug/mL on the Inito monitor and 15 ug/mL on the Mira monitor. However, on both monitors there were two dips in the PDG plateau, illustrating that the prolonged luteinization process may lead to slight changes in the progestation process. PDG declined abruptly (i.e., luteolysis) on the Inito monitor on day 33 and on the Mira monitor on day 34, and the cycle was complete on day 34.

### 3.3. Anovulation

In this anovulatory cycle (Figure 3), no peak day was found on the CBFM, and no rise in LH was found on either the Mira or Inito monitors. On day 20 on the Mira monitor and day 17 on the Inito monitor, there were missing E3G and LH results, but the lack of a PDG rise afterwards suggests that this cycle was not ovulatory. The Mira PDG levels were 7.4, 7.5, and 7.8 ug/mL, which did not reach the plateau levels found in the other two cycles; the Inito PDG levels did not rise above 2.6 ug/mL. In this anovulatory cycle, the luteinization, progestation, and luteolysis processes were absent. 

## 4. Discussion

This case study was a hypothesis-generating exercise to plan larger studies for the quantitative analysis of the luteal phase. The strength of quantitative measurements is in identifying individual differences between women and between cycles, thus personalizing the process of classifying normal and abnormal cycles, rather than assuming that individuals always tend towards a theoretically normal cycle with day 14 as the day of ovulation and a 28-day cycle length. Since there is a complex interplay between hormones in the menstrual cycle, quantitative hormone evaluation by women at home will now empower women to identify their own variations from cycle to cycle, and this can inform their care providers in identifying clinical abnormalities. This personalized approach to the menstrual cycle will also allow health care providers to tailor solutions to each woman’s individual needs, whether to optimize conception or to track her cycles.

The first clinical scenario (a normal cycle, Section 3.1 and Figure 1) demonstrates the three processes outlined in a previous study [9]: (1) luteinization with an LH surge and an initial rise in progesterone, then (2) progestation with a plateau of PDG, and finally (3) luteolysis with a fall in PDG a few days before the next menstrual period.

In the second clinical scenario (prolonged luteinization, Section 3.2 and Figure 2), we demonstrated that a woman could identify broad LH surges, as was demonstrated in previous studies [12,21], and that this may impact the progestation process [9], given that there were dips in the PDG levels for both the Inito and Mira monitors in that cycle. Now that we have these quantitative measures of progesterone, this may enable us to define a luteal phase deficiency more precisely [18,19] and inform the use of supplemental progesterone for luteal phase deficiency or recurrent miscarriages [15]. 

Finally, identifying anovulation with the absence of a progesterone rise (Section 3.3 and Figure 3) will help in cases where women may have missed a narrow LH surge and can confirm that they have not ovulated in a given cycle. Anovulation may occur with polycystic ovarian syndrome, high-level athletes, and women with eating disorders, but can also occur randomly in eumenorrheic women [22]. The analysis of the luteal phase allows us to determine whether ovulation was missed by the woman based on progesterone rises, which can confirm ovulation and, in addition, time progesterone supplementation earlier if needed. Identifying anovulation will help plan for interventions that may help target anovulatory cycles, such as ovarian stimulation [23].

The luteal phase has previously been described as having three distinct processes: first, the luteinization process, in which the corpus luteum is formed based on an interplay between LH and progesterone; second, the progestation process, involving the rise and plateau of progesterone over several days to support a potential pregnancy; and third, the luteolysis process, when progesterone levels fall and the corpus luteum regresses, which then leads to the sloughing of the endometrial lining and menses [9]. With this model in mind, the central process involved in a woman’s cycle is not menstruation but ovulation, and it is ovulation that sets up the three processes of the luteal phase. The quality of ovulation may further be elucidated by these three processes of the luteal phase, as demonstrated in Section 3.2, where prolonged luteinization may impact the progestation process.

The quantitative evaluation of the luteal phase may also have clinical utility in targeting interventions that may assist with premenstrual migraines [24], premenstrual syndrome, premenstrual dysphoric disorder [25,26,27], or, as previously mentioned, luteal phase deficiency to optimize fertility [15].

New fertility monitors that are now available that measure quantitative progesterone levels (Mira, Inito, Proov, and Oova) need to be further validated in clinical studies to ensure the accuracy and reliability of these monitors in delineating the processes of the luteal phase. Based on this proof-of-concept case report, it seems that the Mira and Inito monitors can identify the three processes of the luteal phase. Larger case series and clinical trials could be planned to target clinical interventions in the luteal phase to help optimize fertility and treat premenstrual disorders. 

## Figures and Tables

**Figure 1 medicina-59-00140-f001:**
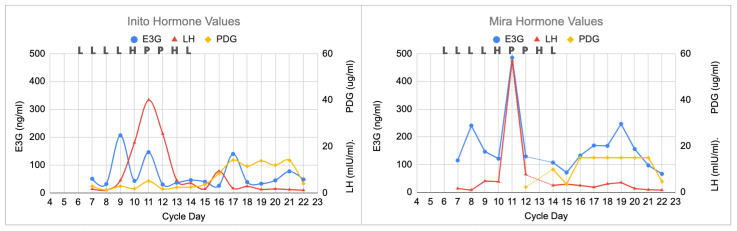
Normal cycle showing all three monitors with agreement on the peak day, with the highest LH value on that day (cycle day 11). The ClearBlue results (L = Low, H = High, and P = Peak) are shown above each graph on the respective days. The estimated day of ovulation was day 12 (day after LH peak). The luteal phase was 13 days (cycle length of 25 days). PDG initially rose on day 16 on the Inito monitor and on day 14 on the Mira monitor.

**Figure 2 medicina-59-00140-f002:**
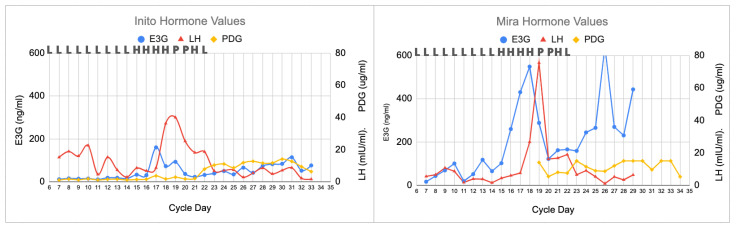
Longer cycle with prolonged luteinization process. All three monitors agreed on the peak day, with the highest LH value on that day (cycle day 19). The ClearBlue results (L = Low, H = High, and P = Peak) are shown above each graph on the respective days. The estimated day of ovulation was day 20 (day after the LH peak). The luteal phase was 14 days (cycle length of 34 days). With both the Inito and Mira monitors, there were two PDG dips on day 25.

**Figure 3 medicina-59-00140-f003:**
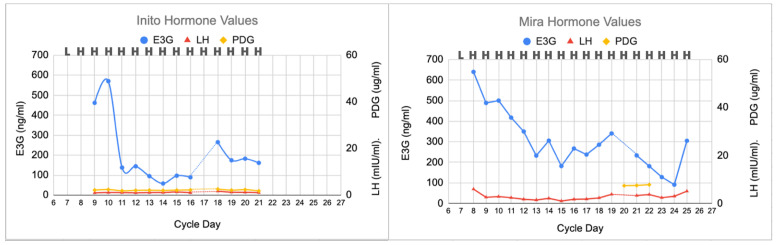
Anovulatory cycle showing no LH surge and no PDG rise. E3G fluctuations were present at the beginning and middle of the cycle, but no luteinization, progestation, or luteolysis processes were observed.

## Data Availability

The raw data are available upon request.

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
