# Peer review of "Using Quantitative Hormonal Fertility Monitors to Evaluate the Luteal Phase: Proof of Concept Case Study"

_medicina, 2023, doi:10.3390/medicina59010140_

Round 1
Reviewer 1 Report
I commend the author for this case report. I think it is well written and deserves publication as it is since it presents a preliminary concept on the use of two novel monitors. I will now suggest two additions which to my opinion will add to the article:
1) I would make a summarizing table comparing Inito and Mira in regard to each of three cycles (to complete the figures side by side) for easy reading: EDO -LH peak day, length of cycle and luteal phase, length of lutenization, day of rise of PDG post LH surge, etc
2) I would like to see more details on anovulatory cyles: how many, duration, any comments why the MIRA would show an ‘ovulatory range’ values for PDG on days 20-22
Author Response
Thank you for your helpful comments.
1) A table of values would be helpful to compare between the two monitors. I would like to know from the editors where these tables of values would best be positioned- in an appendix, beside the Figures 1-3, or separate from the figures? Data tables are attached.
2) The progesterone values in Figure 3 for Mira are NOT in the ovulatory range. These values were 7.4, 7.5 and 7.8 (described in the text), and presumed ovulatory range is >10 as described in the previous cycles with their plateau ranges of 12 (for Inito) and 15 (for Mira).

Reviewer 2 Report
Review from Reviewer 1 is attached.

Author Response
Thank you for your comments.
1) Regarding the ClearBlue thresholds, this is indeed proprietary info, but I included some likely thresholds based on previous study data.
2) Thank you for these very insightful comments about E3G. Because this paper was focused on the luteal phase, and sample size was quite small (case report), we do not feel we could adequately address the E3G follicular phase variability. This will be addressed in a separate larger study in the future.
3) Typos were all corrected.
Reviewer 3 Report
Line 77 importance of identifying abnormal luteal phases and early timing of progesterone supplementation line 116 average peak day ? first day "peak" by CBFM or peak LH by Mira. Specify line 129 The same as in line 116. Here this coincides but I imagine it is not always so. It seems to me that the EDO needs to be detailed a little more, the bibliographical reference alone is not enoughline 133 Specify that broad LH peak is only seen here with Mira
line 146 and 147 day 31 dip with Inito? I would skip cycle day 31Both detect PDG dips on and around day 25
PDG not progesterone. Line 177 prolonged lh peaks described earlier than in bibliography #12: Alliende, Marıa Elena. "Mean versus individual hormonal profiles in the menstrual cycle." Fertility and sterility 78.1 (2002): 90-95. bibliography #12: Direito A, Bailly S, Mariani A, et al. Relationships between the luteinizing hormone surge and other characteristics of the menstrual cycle 248 in normally ovulating women. Fertility and Sterility 2013;99:279–85. Linea178 were dips. I would skip twoLine 188 In addition, progesterone supplementation can be timed early
Author Response
Thank you for these helpful comments.
1) Line 77 was updated as suggested.
2) Line 116 average peak day by CBFM and extra EDO detail was added to the introduction as requested.
3) Line 133 The broad LH peak was seen with Inito as well, so no changes required.
4) Line 146 and 147 as well as in discussion, I excluded comments on the day 31 dip and only referenced the day 25 dip. I corrected progesterone to PDG.
5) The 2002 reference was added to Line 177-179 prolonged lh peaks described earlier than in bibliography.
6) Line 188-190 - this comment was added "In addition, progesterone supplementation can be timed early"
Thank you!